# Tobacco Plant Growth-Promoting and Antifungal Activities of Three Endophytic Yeast Strains

**DOI:** 10.3390/plants11060751

**Published:** 2022-03-11

**Authors:** Mariana Petkova, Slaveya Petrova, Velichka Spasova-Apostolova, Mladen Naydenov

**Affiliations:** 1Department of Microbiology and Environmental Biotechnology, Agricultural University of Plovdiv, 4000 Plovdiv, Bulgaria; sl_petrova@au-plovdiv.bg (S.P.); v_apostolova@au-plovdiv.bg (V.S.-A.); mladen@au-plovdiv.bg (M.N.); 2Department of Ecology and Environmental Conservation, Plovdiv University Paisii Hilendarski, 4000 Plovdiv, Bulgaria; 3Agricultural Academy, Tobacco and Tobacco Products Institute, 4108 Markovo, Bulgaria

**Keywords:** endophytic yeast, antimicrobial activity, indole-3-acetic acid, siderophore, phosphate-solubilization, tobacco, physiological parameters

## Abstract

In this research, the biosynthetic and biocontrol potential of endophytic yeast to improve the growth and development of tobacco has been elucidated. Three yeast strains were enriched and isolated from different plant tissues. Partial sequence analysis of ITS5-5.8-ITS4 region of the nuclear ribosomal DNA with universal primers identified YD5, YE1, and YSW1 as *Saccharomyces cerevisiae* (*S. cerevisiae*), *Zygosaccharomyces bailii* (*Z. bailii*), and *Saccharomyces kudriavzevii* (*S. kudriavzevii*), respectively. When cultivated in a medium supplemented with 0.1% L-tryptophan, isolated yeast strains produced indole-3-acetic acid (IAA). The capacities of those strains to improve the mobility of phosphorus and synthesize siderophores has been proven. Their antimicrobial activities against several Solanaceae plant pathogenic fungi (*Alternaria solani pathovar. tobacco, Rhizoctonia solani,* and *Fusarium solani pathovar. phaseoli*) were determined. *S. cerevisiae* YD5, *Z. bailii* YE1, and *S. kudriavzevii* YSW1 inhibited the growth of all tested pathogens. Yeast strains were tested for endophytic colonization of tobacco by two different inoculation methods: soil drench (SD) and leaf spraying (LS). To establish colonization in the various tissues of tested tobacco (*Nicotiana tabaccum* L.) plants, samples were taken on the seventh, fourteenth, and twenty-first days after treatment (DAT), and explants were inoculated on yeast malt agar (YMA). Both techniques of inoculation showed a high frequency of colonization from 83.33% to 100%. To determine the effectiveness of the microbial endophytes, their effect on some physiological processes in the plant were analyzed, such as photosynthesis, stomatal conductivity, and transpiration intensity. The effect of single and double treatment with yeast inoculum on the development and biochemical parameters of tobacco was reported. Plants have the ability of structural and functional adaptation to stress effects of different natures. All treated plants had a higher content of photosynthetic pigments compared to the control. Photosynthesis is probably more intense, and growth stimulation has been observed. The chlorophyll a/b ratio remained similar, and the total chlorophyll/carotenoid ratio slightly increased as a result of elevated chlorophyll levels. The most significant stimulating effect was recorded in tobacco plants treated by foliar spraying with *Z. bailii* YE1 and *S. cerevisiae* YD5. In contrast, *S. kudriavzevii* YSW1 had a better effect when applied as a soil drench. Thus, *S. cerevisiae* YD5, *Z. bailii* YE1, and *S. kudriavzevii* YSW1 have a high potential to be used as a biocontrol agents in organic agriculture.

## 1. Introduction

In recent years, increasing attention has been paid to organic farming, which has become a priority in agriculture worldwide. There are two main problems to be solved to interchange the common agricultural practices with organic ones: pest control and high yields recovery without using chemicals. These problems can be addressed using different organisms (bioagents) with higher antifungal or growth-promoting potential. The importance of the soil rhizosphere enrichment with beneficial microorganisms to boost the conditions of root nutrition has led to the appliance of microbial fertilizers [1]. They show significant advantages over artificial synthetic fertilizers, namely, they synthesize physiologically active substances (vitamins, auxins, etc.) and provide them to the plant, increase the productivity of photosynthesis, enhance enzymatic processes in plants and improve water regime, induce the activity of other soil microorganisms, act as antagonists of phytopathogenic microorganisms, and increase the protective functions of plants [2,3,4]. Extensive research has led to the development of various microbial-based products that can be used in the biocontrol of plants [5]. Yeasts are broad-distributed, and they live in association with other microorganisms that are common populations of soil, vegetation, and aqueous environments. One of the foremost studied yeast species is *Saccharomyces cerevisiae* due to the growing interest in industrial processes [6,7]. Endophytic yeasts have a unicellular part of their lifecycle and usually reside within plant tissues and do not cause damage to their hosts [8]. The endophytes can be locally or systemically distributed in various plants where they are protected from biotic and abiotic stress [9]. Endophytic yeast can ecologically adapt to host plants and overcome plant defence responses [10]. Microbial distribution and species diversity in the tissues of fruits, seeds, and roots are studied. Usual methods are to directly observe endophytic yeast within the plant tissues using a microscopic technique. Isaeva et al. (2010) [11] found that yeast cells are most frequently found in the intercellular space in plant tissues. In yeast, identification is based not only on their morphology but also their biochemical and physiological characteristics. The most up-to-date research is the development of methods that involve the routine employment of samples based on DNA sequences unique to the species. Recent studies have found that endophytic yeast can be isolated from various parts of plants, such as *Williopsis saturnus* from corn root [12], *Candida guilliermondii* from the stem and tomato leaves [13], and *Rhodotorula graminis* strain WP1 from *Populus trichocarpa* [14]. The endophyte’s yeast–host relationship can provide growth benefits by protecting it from insect attack and disease [15]. Therefore, it can be used successfully as a biocontrol agent. Several yeasts are known to produce mycocins (extracellular proteins) involved in inhibition of *β*-glucan synthesis in the cell wall of sensitive strains, which cause the death of the other species of yeast, moulds, and bacteria [16]. Yeast is involved in the process of dissolving soil-insoluble phosphates, transforming complex organic biomass into mineral compounds used by plants, and the synthesis of growth factors such as amino acids, vitamins, and enzymes [17]. There have been an increased number of studies concerning the isolation of endophytic yeasts that stimulate plant growth [18]. Plant growth-promoting yeasts can increase yield and reduce pathogen infection as well as biotic and abiotic stress [19]. The direct effect of stimulation of plant growth is achieved by the production of phytohormones [18]. Indole 3–acetic acid (IAA) is the most common plant auxin and regulates plant growth and development [20]. The role of microbial IAA in plant–microbe interactions has recently received increased attention as a signaling molecule in microbe–plant interactions [21]. Yeast may be possible for pesticide degradation and its remediation [22]. *Saccharomyces cerevisiae* can act as a PGP and a biocontrol agent for sustainable agriculture in unadorned conditions [6]. Baker yeast can be also involved in stress tolerance [23]. *Saccharomyces cerevisiae* was reported to have the potential to inhibit plant pathogens [24]. Among fungal pathogens, chitin-containing pathogens such as *Fusarium solani,* *Rhizoctonia solani*, and *Alternaria solani* *induce* damage to tobacco such as root and stem rot and leave brown spots [25].

All these new insights into the plant–microbe relationships create a broad field of investigations for their potential in the industrial crops growing. Yeast application could lead to effective results in the biological control of economically important diseases in many industrial crops [26]. The antimicrobial properties of yeast are mainly in arrears to competition for nutrients, changes in the environment, as a result, organic acid production, increased concentrations of ethanol, and the release of antimicrobial compounds like killer toxins (mycocins) and siderophores [27,28,29]. Mycocins are extracellular proteins or glycoproteins that disrupt the function of the cell membrane in sensitive microbial species that carry receptors for these walls [29]. Yeast is a good producer of hydrolytic enzymes that destroy other microbial cell walls [30,31]. Some yeast species are also able to produce siderophore compounds (iron (III) ion compounds) that inhibit the growth of fungal phytopathogens [32,33]. The iron deficit in plants reduces metabolic activity and biomass and leads to chlorosis. The soil fungi and plant endophytes synthesize two types of siderophores-hydroxamate and carboxylate (coprogens, fusarinines, and ferrichrome) [34,35]. Microbial siderophore synthesizers have the potential application in agriculture for the depolymerization of the wood cell walls [36]. Endophytic fungi *Candida aloifolium* was reported to produce the siderophore on Chrome Azurol S (CAS) agar based on their affinity towards ferric iron. Some data are already published for *C. aloifolium*, which is well-known as a bio-agent for the control of fungal pathogens after harvest [37].

The present study aims to evaluate the effect of three endophytic yeast strains on tobacco plants physiology and development to demonstrate PGP activity and their possible application in organic farming. Isolation of plant growth-promoting microorganisms (PGPR) including IAA-producers and siderophore-producing yeast has been performed using different host plant species, and their genetic, biochemical, and metabolic characteristics have been studied. The effect of single and duplicate treatment with biocontrol yeast strains on tobacco growth and development is described for the first time.

## 2. Materials and Methods

### 2.1. Isolation and Molecular Identification of Yeast

Yeast was isolated from surface-sterilized leaves of strawberry (*Fragaria × ananassa*) and wheat seeds (*Triticum aestivum*) grown on the training-experimental field of the Agricultural University of Plovdiv, spreading on 185 ha around the city of Plovdiv, South Bulgaria. The plant samples were washed with running tap water to remove dust and debris adhering to them [38]. The surface sterilization to remove the adhering microorganisms was performed by immersion in a 3% sodium hypochlorite (commercially available) solution for three minutes. Then, they were rinsed with 70% ethanol for a minute. The seed and leaves finally were rinsed with deionized sterile distilled water to remove the superficial chemical agents. The sterilized seed and leaf explants were cultured in Petri dishes containing YMA (Himedia Laboratories Pvt., Mumbai, India) supplemented with 100 μg/mL of chloramphenicol. The Petri dishes were sealed with parafilm and incubated at 27 °C for 5 days under dark conditions and monitored every day. For the control of the sterilization of the explants, dsH_2_O from the final washing step was antiseptically dripped onto an antibiotic-free YMA. No yeast colonies were detected from the final washing water used for sterilization of the explants. Yeast colonies, isolated from each plant explant, were subcultured on separate YMA plates at room temperature, morphologically analyzed using microscopy, and then subjected to molecular identification.

For molecular identification, DNA was isolated with a HiPurA fungal DNA purification kit (Himedia, India) according to the manufacturer’s instructions. Control of the purity and concentrations of genomic DNA was performed by agarose gel electrophoresis. Yeast identification sequences of large subunit (LSU) ribosomal RNA (rRNA) were determined from polymerase chain reaction (PCR) products from the genomic DNA extracted from yeast cells. LSU rRNA was amplified using a PCR with the universal primers with 2 µL each of 10 µm internal transcribed spacer ITS-5 and NL-4 and 0.5 µL Taq polymerase (5 U/µL, Canvax, Spain) [39]. The parameters of thermal cycling were initial denaturation at 96 °C for 2 min, followed by 30 cycles consisting of denaturation at 96 °C for 1 min, annealing at 55 °C for 45 s, and extension at 72 °C for 2 min, followed by a final extension at 72 °C for 5 min. Samples were sent to Macrogen (Seoul, South Korea), for DNA sequencing. A BLAST search of nucleotide sequences was performed using the platform of the National Centre for Biotechnology Information (http://www.ncbi.nlm.nih.gov (accessed on 17 December 2021 and the day of the last accession was 24 January 2022). Yeast identification followed the guideline of Kurtzman (2011) [40], who reported that yeast strains with 0–3 nucleotide differences are conspecific or sister species, and different species were identified if they had six nucleotide substitutions (Appendix A).

### 2.2. Biosynthetic Potential of the Studied Strains

#### 2.2.1. Production of Indole-3-Acetic Acid

Quantification of indole-3-acetic acid was made using the Salkowski reagent [41]. Yeast strains were grown in a test tube in yeast extract agar (Merck KGaA, Darmstadt, Germany) with or without 0.1% (*w/v*) L-tryptophan and incubated in the dark on a shaker at 30 °C and 150 rpm/min for 5 days [42]. Produced IAA was quantified using calibration curve (Appendix A). One milliliter of the cells was pelleted by centrifuging at 3000 rpm for 5 min, and 0.5 mL of the supernatant was mixed with 0.5 mL of Salkowski reagent (2 mL of 0.5 M iron (III) chloride and 98 mL of 35% perchloric acid) according to Sun et al., 2014. After 30 min, the development of an orange-red color was quantified using a spectrophotometer (Shimadzu UV-1800, Thermo Fisher Scientific, Waltham, MA, USA) at 530 nm. IAA concentration was detected by the establishment of a calibration curve using different concentrations of pure IAA (Appendix A). Statistical analysis data are expressed as mean standard deviation. Student t-tests were used for the determination of differences between groups and analyses of variance. *p* < 0.05 was considered statistically significant. 

#### 2.2.2. Assay for NH_3_ Production

Yeast isolates were tested for the production of ammonia in peptone water. Isolated strains were inoculated into 10 mL peptone water and incubated for 48 h at 30 °C. After the incubation, 0.5 mL Nessler’s reagent was added to each tube. The presence of a brown-to-orange color was considered as a positive test for ammonia production [43].

#### 2.2.3. Screening of Phosphate Solubilizing Yeasts

In order to establish phosphate-solubilizing yeasts, each of the isolated yeast strains were inoculated on Pikovskaya’s agar (PVK, yeast extract 0.5 g/L, dextrose 10 g/L, Ca_3_(PO)_2_ 5 g/L, (NH_4_)_2_SO_4_ 0.5 g/L, KCl 0.2 g/L, MgSO_4_.7H_2_O 0.1 g/L, MnSO_4_.H_2_O 0.0001 g/L, FeSO_4_.7H_2_O 0.0001 g/L, and agar 15 g/L, pH 7.0) (Himedia, India) [44]. The plates were incubated at 25 ± 2 °C for 5 days. Yeasts capable of producing a clear zone due to solubilization were selected as potential phosphate solubilizers and used for further studies.

#### 2.2.4. Screening for Siderophore Production and Proteolytic Activity

To detect the production of siderophore from yeast, a medium with 2% yeast malts agar (Himedia, India) was used, in which the yeast developed well, and then CAS according to [45] was added. After inoculation of the medium, the yeast was incubated at 25 °C for 7 days in the dark. Yeast growth and medium color change (from blue to purple or yellow) were monitored. Development of a yellow-orange halo around the colony was considered a positive result for siderophore production ability.

Proteolytic activity (casein degradation) was determined from a clear zone in skimmed milk agar (Himedia, India). The agar plates were prepared and spot inoculated with tested yeast strains and incubated at 30 °C for 5 days. A halo zone around the colony was measured and accepted as positive for cell-wall-degrading enzyme production.

### 2.3. Antimicrobial Activity against Fungi of the Genus Fusarium, Rhizoctonia, and Alternaria

The studied yeast strains were tested to determine their antifungal action against *Alternaria solani pathovar. tobacco, Rhizoctonia solani,* and *Fusarium solani pathovar. phaseoli*. Pathogenic fungi are provided by the collection of the Department of Phytopathology, Agricultural University of Plovdiv. Pathogenic fungi were cultured at 25 °C on yeast extract medium (5 g yeast extract, 10 g glucose, and 20 g agar) for 7 days. Their conidia were then collected by washing with cold sterile distilled water and used to prepare an inoculum at a concentration of 1 × 10^4^ spores/mL. Tests for antifungal activity were performed by agar diffusion [46]. 

### 2.4. Monitoring the Effect of Yeast Colonization on the Development of Tobacco in Pot Experiments

The experimental design was set up to track and assess the effect of three factors: (1) yeast strain: *S. cerevisiae* YD5, *Z. bailii* YE1, and *S. kudriavzevii* YSW1; (2) type of application: soil drench or leaf spraying; (3) several applications: single or double. Tobacco seedlings, (*Nicotiana tabacum*, L.) variety Krumovgrad 58, were taken from flower beds at the Institute of Tobacco and Tobacco Products (Markovo Village, Bulgaria). A total of 55 tobacco plants (BBCH phase 45 days old tobacco seedlings) were inoculated individually by soil or leaf treatment, and five plants were used as a control. Pots with a capacity of 3.5 litres were used for the experiment. The soil application was made by pipetting near the root and the leaf spraying was made on all leaves with a suspension at a concentration of 1 × 10^4^ yeast cells, separately for each strain. Control plants were not treated with yeast suspension. On the fourteenth day after the first inoculation, an additional second treatment was performed for half of the tobacco plants.

Biometric measurements have been made regularly to track the effect of yeast colonization on tobacco plant development. The root length and stem height were measured before transplanting into pots, and on the 7th, 14th, 21st, 28th days after treatment (DAT). The number of leaves, leaf and root biomass have been also observed during the experimental period.

#### 2.4.1. Plant Colonization by Endophytic Yeast Strains

On the fourteenth DAT, some samples of treated plants were removed to determine the presence of yeast in them by inoculating explants from leaves, stems, and roots on yeast extract agar (Merck, Germany). Control plants were also taken. The plants were removed from the soil and washed with dH_2_O. Surface sterilization of leaves, stems, and roots was performed before the introduction of in vitro explants for 3 min in 0.008% Tween 80 *w/v*, 3 min in sodium hypochlorite NaOCl solution, 1 min in 70% ethanol, and three times rinsing with sterile dH_2_O for 50 s [47]. Six leaf discs with a size of approximately 1 cm^3^ were seeded on a nutrient medium with added antibiotics of 100 µg/mL ampicillin, 100 µg/mL streptomycin, and 100 µg/mL tetracycline. The presence of yeast was reported on day 5 after incubation at 27 °C in the dark.

The isolation frequency (IF) of the colonization of tobacco by yeast strains was calculated by the following formula published by Petrini and Fisher (1986) [48].

#### 2.4.2. Biometrical Analysis of Tobacco Plants after the First and the Second Treatment with Yeast Strains (*S. cerevisiae* YD5, *Z. bailii* YE1, and *S. kudriavzevii* YSW1)

The effects of tested endophytic yeast strains on the growth and development of tobacco plants and on their adaptation were determined in colonized plants. Biometrical analysis by the important traits as stem height, root and leaf length, and leaves’ biomass of the aboveground part of tobacco was taken weekly. The statistical data were processed with MS Excel 2010. Measurements were conducted in five analytical and five biological replicates.

#### 2.4.3. Monitoring Changes in Parameters of Photosynthesis and Transpiration

Pigment analysis followed the method published by Schlyk (1971). The extraction of photosynthetic pigments was performed with 90% acetone, and spectrophotometrical measurements were performed at 440.5 nm for carotenoids, 644 nm for chlorophyll b, and 662 nm for chlorophyll a [49]. The analyses were conducted in three replications. Concentrations of chlorophyll a, chlorophyll b, total chlorophyll, and carotenoids are calculated for each sample and presented in mg g^−1^ fresh weight as arithmetic mean ± SD.

Physiological measurements were performed on three fully developed, intact leaves of the same physiological age. For this purpose, a portable photosynthetic system Q-box CO650-Plant CO_2_ Analysis Package (Qubit Systems Inc., Kingston, Canada) was used. The intensity of photosynthesis (A, μmol m^−2^ s^−1^), stomatal conductance (Gs, μmol m^−2^ s^−1^), and intensity of transpiration (E, μmol m^−2^ s^−1^) was monitored, and the results are presented as arithmetic mean± SD.

#### 2.4.4. Statistical Evaluation

Statistica 7.0 software was used for the statistical evaluation [50]. Raw data were processed by the t-test (dependent samples), and cluster analysis was performed for the grouping of the studied sampling sites based on studied biometrical and physiological parameters. Relationships between the studied parameters in collected leaf samples were tested using Pearson correlation coefficients. All the analyses were significant at *p* < 0.05.

## 3. Results and Discussion

### 3.1. Isolation and Molecular Identification of Yeast

A total of three morphologically distinct yeast isolates were selected and evaluated for their PGP and antagonistic traits. To achieve a more complete and correct species identification of endophytic isolates, one of the widely accepted methods of species identification by sequencing the LSU rRNA gene and comparing the resulting sequences with global databases was used. YD5 was identified as *S. cerevisiae,* YE1 as *Z. bailii*, and YSW1 *S. kudriavzevii,* respectively. The nucleotide sequences obtained in this work were deposited in GenBank under accession numbers that are given in Table 1. It should also be noted that, in the last few years, the NCBI database has been significantly enriched with newly identified species for the *Saccharomycota* yeast group.

Research on endophytic bacteria and filamentous fungi is more in-depth than research on the role, biology, and genetics of yeast [51]. Recent studies are encouraging the potential of endophytic yeast for industrial and agricultural applications, providing strong incentives for more in-depth analysis [52]. They can have significant advantages over bacterial and fungal endophytes since they can be easily cultivated, stored for a long time, and safely applied in different cultures. Although more research is needed and especially field trials to assess their potential, their use can be a biological way to reduce the cost of fertilization and artificial irrigation in agriculture, as well as potentially increase yields. Their application seems particularly promising in the field of bioremediation of soils contaminated with heavy metals or as biocontrol agents for plant protection against pathogens [53,54]. However, many aspects of the biology of endophytic yeast still need to be clarified, especially when it comes to the mechanisms by which yeast can colonize plants. Several studies have shown that the degree of root colonization is crucial for stimulating growth and increasing yields and tolerance to biotic and abiotic stress [55,56].

Most of our knowledge of endophyte characteristics, increased endophyte growth, bacterial endophyte study, and comparisons can be useful in endophytic brain study. This includes the production of phytohormones, reducing stress, protecting against pathogens, and increasing the intake of nutrients from the plant.

The study of endophytic bacteria is more in-depth; endophytic biology and the ecology of yeast remain poorly understood. This stems from their similar way of life in terms of their ecological niches, as well as their physiology, including their unicellular existence, adapted to aquatic environments, rapid growth rates, and the importance of biofilms in their life cycle.

Most of our knowledge about endophyte characteristics and mode of action came from elucidating bacterial endophytes. This includes the production of phytohormones, reducing stress, protecting against pathogens, and increasing the intake of nutrients from the plant.

Yeasts applied to plants seem to spread systematically, unlike some fungi, and can be easily cultivated in a manner similar to bacteria. They also have advantages over bacteria, including their ability to freeze-dry more efficiently than bacteria and therefore be easier to distribute for agricultural use [57]. In addition, there is evidence to suggest the possibility of forming a mixed biofilm containing both bacteria and yeast [58]. The production of plant hormones provides a direct method of promoting plant growth by endophytes. Auxins and gibberellins stimulate plant growth by supporting root growth and stem elongation, as well as, more broadly, cell proliferation and elongation. In particular, the production of indole-3-acetic acid (IAA) by endophytic yeast has been widely reported in the literature [12,59].

### 3.2. Biosynthetic Potential of the Studied Strains

#### 3.2.1. IAA Production

The amino acid tryptophan is a prominent precursor of IAA. Phytohormone IAA is the most common naturally occurring and most methodically examined plant growth regulator. The current experiment showed that the concentration of IAA production was strain-specific. In the current study, all three yeasts were able to produce IAA. The tested yeasts have a medium level of IAA synthesis ranging from 0.463 ± 0.024 mg/L by *Z. bailii* YE1 and 0.512 ± 0.017 mg/L by *S. cerevisiae* YD5 to 0.625 ± 0.012 mg/L by *S. kudriavzevii* YSW1 strain when there is 0.1% L-tryptophan in the medium (Figure 1). *S. kudriavzevii* YSW1 produces more indole acetic acid compared to *S. cerevisiae* YD5 and *Z. bailii* YE1. According to Nutaratat et al. 2015, the red yeast *Rhodosporidium paludigenum* was the robust producer of IAA from tryptophan in a liquid medium with the level of 1.623 mg/L [60]. In yeast, IAA has been proposed as a metabolite of tryptophan [61], which was confirmed in later studies in various yeasts such as *Saccharomyces uvarum* and *Saccharomyces carlsbergensis* [62].

IAA is the first plant hormone to be discovered but the biosynthesis of indole-3-acetic acid in yeast is not well-described. Previous studies suggest that there are four pathways of IAA biosynthesis in plants and bacteria [20]. The synthesis of microbial phytohormones from yeast is associated with signaling changes in the root and stimulating plant growth. Several studies have observed a correlation between plant growth and the concentration of hormones measured in the culture medium or colonized plant tissues during experiments and in situ [63]. Most of the research in the past has been focused on the diversity and ecology of yeasts rather than their applications [52].

Tryptophan is not biosynthesized by all microorganisms, and these microorganisms rely on their plant hosts or surrounding microbial sources [64]. Hernández-Fernández et al. 2021 discovered that the first step is the conversion of tryptophan to IPA by aminotransferase activity, then IPA is carboxylated to indole-3-acetaldehyde (IAAld) by indole-3-pyruvate decarboxylase (IPDC) activity, and the last step is the oxidation of IAAld to IAA [65]. Alternatively, IPA can also be converted directly to IAA by indole-3-pyruvate monooxygenase. IAA production is regulated by the strain used, its growth phase, the precursor concentration, and medium components, and each yeast strain has a characteristic biosynthesis pathway [66]. Moreover, the ability to convert tryptophan to IAA by these microorganisms can also benefit the plant, which may be seen as an example of mutualism [18].

#### 3.2.2. NH_3_ Production

Plant roots are colonized by a diversified population of endophytic microorganisms including beneficial bacteria and fungi. Positive effects from yeast application could be due to the provision of soluble nutrients that can contribute directly to plant growth and act as biostimulants. Ammonia production is a very important PGP characteristic of microorganisms. In the current experiments, *S. cerevisiae* YD5, *Z. bailii* YE1, and *S. kudriavzevii* YSW1 exhibited strong ammonia production abilities (Table 1).

#### 3.2.3. Phosphate Solubilization

Acid phosphatases and phytases synthesized by rhizosphere microorganisms are involved in the organic dissolution of phosphorus in soil [67]. Phosphorus is one of the sources of energy, which often limits plant growth due to its poor solubility and fixation in the soil. The transformation of insoluble and fixed forms of phosphorus into soluble forms is an important aspect of increasing phosphorus in soil [68]. Microorganisms play a critical role in insolubilization and mineralization in inorganic and organic soils [18]. The phosphate-solubilizing index (PSI) of the studied yeast strains was determined by a qualitative method of analysis using Pikovskaya agar. *Z. bailii* YE1 had the strongest solubilizing activity of 27.6 ± 0.19 (*p* < 0.05). The other two strains were found to have similar moderate phosphate-solubilizing activity in PVK medium measured as the PSI for *S. cerevisiae* YD5–16.6 ± 0.142 and *S. kudriavzevii* YSW1–16.0 ± 0.138 (Table 1).

These phosphate-solubilizing microorganisms transform insoluble phosphate into a soluble form through the production of organic acids, phosphatases, or other complex agents [69]. Singh and Satyanarayana, 2011 reported that the predominant forms of organic phosphorus are phytates, which make up 60% of soil organic phosphorus [70]. To be absorbed by plants, phytates must first be dephosphorylated with phosphatases [71]. Therefore, the application of phosphate solubilizing microorganisms to fields has been reported to increase crop yield. Most of the extensively examined microbial mediated species has been using bacteria and filamentous fungi [72]. *Yarrowia lipolytica* has been reported to solubilize inorganic phosphate. Endophytic yeast has been reported to play a role in P-solubilization by several mechanisms, such as lowering the pH by acid production, iron chelation, and exchange reactions in the growth environment [52]. Application of these endophytes in agriculture could reduce inputs of water and fertilizer. Previous studies showed that growth promotion by endophytic yeasts focused on their potential to be used for bio-augmented phytoremediation of heavy metals [73]. Heavy-metal-contaminated soils are significant health risks, and current technologies for remediation are expensive and insufficient [74].

#### 3.2.4. Siderophore Production and Proteolytic Activity

Yeast improves plant growth through components that induce resistance to environmental stress, leading to the production of antifungal compounds such as siderophores. The production of siderophores is associated with the formation of soluble Fe^3+^ complexes, which are involved in active membrane transport and can be received by plant cells [75]. The synthesis of siderophores from yeast in the rhizosphere leads to impaired access of iron to harmful microflora and is reported as a significant PGP characteristic [76]. In the present study, the ability of strains to produce siderophores was determined on liquid and solid media after the addition of CAS according to Schwyn and Neilands (1987) [45]. The synthesis of siderophores from yeast strains was established by a change in the color of the nutrient medium (from blue to red-orange or yellow). Interestingly, strain *Z. bailii* YE1 and *S. cerevisiae* YD5 showed higher activity on the solid medium compared to the liquid medium with CAS. *S. kudriavzevii* YSW1 actively produce siderophore when cultivated in the liquid medium but lack the activity in the solid medium (Table 1). They were selected to treat plants and to monitor the effect on the growth and development of tobacco. Siderophore production is a common trait of endophytic yeasts that have been considerably reviewed and found in rice and sugar cane leaf endophytes [77].

Microorganisms have a different mechanism of pathogen inhibition through the production of cell-wall-depredating proteolytic enzymes. The activity of extracellular protease was detected in vitro by clear zones on skim milk. Results indicated that all the strains produced protease. The highest protease activity of the 35 mm zone was measured when YD5 was applied, followed by a 32.5 mm zone by the YWS1 stain. Compared to them, YE1 showed a decrease in the proteolytic capacity, with 46.8% compared to YD5 and 50.4% compared to YSW1. 

#### 3.2.5. Antifungal Activities

Another important trait of PGPR, which may indirectly influence plant growth, is the production of siderophores [78]. They bind to the available form of iron Fe^3+^ in the rhizosphere, thus making it unavailable to the phytopathogens and protecting the plant health [79,80]. In the present investigation, the siderophores produced by endophytic yeast *S. cerevisiae* YD5, *Z. bailii* YE1, and *S. kudriavzevii* YSW1 exhibited antifungal activity against the potent tobacco pathogens such as *Alternaria solani pathovar. tobacco, Rhizoctonia solani,* and *Fusarium solani pathovar. phaseoli*. The siderophore produced by *S. cerevisiae* YD5 recorded the strongest inhibition against *F. solani*, followed by *S. kudravzevii* YSW1. *Z. bailii* YE1 suppressed the growth of all the tested fungal pathogens and exhibited similar activities against them. It has been reported that *Zygosaccharomyces* spp. are very tolerant to high sugar conditions and resistant to sulphite and sorbate at levels that are inhibitory to other yeasts and bacteria [81]. The present finding proves that yeast siderophores are potent agents that can be used against *Solanaceous* plant pathogens. These results are in accordance with Leong (1986) and revealed that microorganisms produced siderophores during iron-limiting conditions sequester iron (III), thus making it unavailable to the pathogen [80]. In 2007, Miethke reported that the external application of siderophores utilizes iron, thereby depleting the availability of iron to the pathogen, hence enabling the killing of the plant pathogen [79]. Earlier findings have reported the use of siderophores in controlling a few pathogenic fungi such as *Pythium ultimum*, *Sclerotinia sclerotiorum,* and *Phytophthora parasitica*, causing diseases in plants [82]. A large group of active yeast strains have been described as biocontrol agents due to their combined ability to produce various antifungal metabolites [83,84]. The antifungal activity of the tested yeast strains has been studied, and the inhibitory effects are presented in Figure 2.

### 3.3. Monitoring the Effect of the Studied Strains on the Development of Tobacco in Pot Experiments

#### 3.3.1. Evaluation of the Ability of Endophytic Yeast Strains to Colonize Tobacco

Published information on endophytic colonization by different yeast species is limited. Joubert and Doty (2018) suggested that yeasts are able to move around their environment and colonize different plant hosts. The authors give a thought-provoking hypothesis to explain yeast colonization as the role of the production of IAA. This hormone affects plant roots by inhibiting the differentiation of plant root cells and promoting root elongation. Although this has not been shown directly, as plant-associated fungi, endophytic yeasts likely have the enzymatic capabilities to degrade plant cell walls [52]. The current study followed up the specificity of yeast to colonize a particular plant species. Tobacco belongs to the *Solanaceae* family, and studied yeast strains are isolated mainly from wheat and strawberry leaves. To establish this ability, the colonization of various tissues in the root, stem, and leaves of tobacco was observed.

The property of synthesizing indole acetic acid is considered an effective tool for screening beneficial microorganisms [43]. In the current experiment, two different inoculation methods have been used: soil drench (SD) and leaf spraying (LS). In the early days of the experiment, when using the technique of soil inoculation, the highest frequency of colonization from 75% was reported for *S. kudriavzevii* YSW1, while *Z. bailii* YE1 has a lower degree of colonization (50%) of tobacco roots (Table 2). The colonization of leaves ranged from 15% after inoculation with *Z. bailii* YE1 to 45% with *S. kudriavzevii* YSW1 strain. Soil drench application of *S. kudriavzevii* YSW1 has the highest colonization rate of 50% on stems of treated tobacco plants. After processing the data from 14th and 21th DAT, the higher percentage of frequency in leaves and stems is impressive, especially in the leaf-sprayed experimental plants. Yeast migration both downstream and upstream was proved up to 21 DAT where the percentages of colonization in different tissues of soil-treated and leaf-treated tobacco plants were quite similar. With a proven difference on the 21 DAT stood out the roots, stems, and leaves of leaf and soil-treated tobacco plants with the yeast *S. cerevisiae* YD5 strain.

#### 3.3.2. Monitoring the Influence of Endophytic Yeast on the Growth Parameters of Tobacco Plants

The results show that the largest growth of the stem was observed in soil-treated plants with *S. kudriavzevii* YSW1. On the 7 DAT, close values were reported in soil and foliar treatment of the growth of the aboveground part of the plants (Table 3). In contrast to these data, on the 14 days until 28 days, the plants with foliar-applied *S. cerevisiae* YD5, *Z. bailii* YE1, and *S. kudriavzevii* YSW1 separately have almost twice the higher stem and produce bigger biomass. From the reported data on stem growth measured on 28 DAT, it is noticeable that soil-treated plants with *S. kudriavzevii* YSW1 have better values than foliar-treated plants with the same strain. The major increase in the stem was detected in the leaf-sprayed with *Z. bailii* YE1 of 35.0 ± 6.58 to 37.5 ± 5.64 cm, followed by the leaf-sprayed with *S. cerevisiae* YD5-28.4 ± 5.32 cm. The longest root length 8.8 ± 3.33 cm was reported in soil-drench treated (SD II) tobacco plants with *S. cerevisiae* YD5 followed by the 8.7 ± 2.92 cm of *S. kudriavzevii* YSW1. The longest leaf length of 20.7 ± 5.11 cm was measured with SD-II treated plants with *S. kudriavzevii* YSW1 followed by 20.0 ± 3.28 cm when plants was single treated with *Z. bailii* YE1 by leaf spaying. The highest value of leaf biomass was found with the same strains and methods of treatment.

After the first treatment, the highest values in leaf-sprayed plants with *Z. bailii YE1* and *S. cerevisiae YD5* have been observed (Table 4). It was observed that plants treated with *S. kudriavzevii YSW1* showed better values in soil treatment, and plants treated with *S. cerevisiae YD5* and especially *Z. bailii YE1* had a better effect in foliar treatment and had higher values than control plants. 

After the second treatment, the soil-drench plants with *S. kudriavzevii YSW1* and leaf-sprayed plants with *S. cerevisiae* YD5 and *S. kudriavzevii* YSW1 show higher growth rates of the aboveground part compared to a single treatment (Table 4). In all other experimental variants, the values of growth of the aboveground parts reported after the second inoculation was lower compare with the single treatment.

#### 3.3.3. Effects of Endophytic Yeast on Physiological Parameters of Tobacco Plants

Many authors have shown that the significant changes in photosynthetic pigments content serve as an adaptive strategy towards a different type of exogenous or endogenous stress, so we aimed to assess the impact of endophytic yeast colonization on plant physiology and development [85,86]. Our data revealed that all applied yeast strains in all types of treatments significantly increased the concentration of photosynthetic pigments in tobacco leaves (Table 5 and Table 6).

An exception was found only for the *S. kudriavzevii YSW1* strain, where the second leaf treatment led to the lowest values of both pigments content and pigments ratios (79–98% against the control). The increased synthesis of chlorophyll b was more pronounced in all experimental tobacco plants, followed by that of carotenoids. Generally, the chl b is considered to be more susceptible to abiotic influences, and during the process of chlorophyll degradation, chl b is converted into chl a [87]. On the other hand, the carotenoids perform many important physiological functions in plants: influencing development and adaptation mechanisms, suggesting coordination of their synthesis in different physiological processes, but mostly serving as antioxidants against endogenous and exogenous oxidative stress occurring when the number of oxidants in the body or cell exceeds that of antioxidants [88]. Nevertheless, the chlorophyll a/b ratio and the total chlorophyll/carotenoids ratio maintained similar levels when compared to the control tobacco plants (83–92% and 90–106%, respectively).

Based on these findings, it could be concluded that there is no inhibitory impact on plant health status and physiology caused by the endophytic yeast strains colonization. Furthermore, they enhance the photosynthetic potential as well as the plant productivity, which is also well-demonstrated by data for photosynthesis rate monitoring (Table 7 and Table 8). 

Periodical measurements (7th, 14th, 21st, and 28th day after inoculation with yeast strains) on the intensity of photosynthesis, transpiration, and stomatal conductivity of experimental tobacco plants indicated that mutual plant–yeast coexistence causes no stress for the first 7 days after treatment. The results obtained correspond to the findings of many authors that the adaptation of the plants to extreme values of any factor includes two basic reactions, to avoid stress and/or stress resistance, and has three main stages. The first stage involves primary stress reactions, the second stage is connected to the adaptive responses to stress due to the antioxidant defensive system of plant organisms, and the third one is related to weight loss and plant death [89]. The most significant positive impact on studied physiological traits exerted the *Z. bailii* YE1 strain, followed by *S. kudriavzevii* YSW1 and *S. cerevisiae* YD5 strains. 

When regarding the effect of leaf spraying versus root drench and the effect of single treatment versus duplicate treatment, the statistical evaluation of all studied biometric and physiological parameters highlighted that, for the *Z. bailii* YE1 strain, the most effective is twofold foliar inoculation, while for *S. cerevisiae* YD5, it is single soil and single leaf treatment, and for the *S. kudriavzevii* YSW1 strain, it is twofold soil treatment (*p* < 0.05). Cluster analysis based on studied parameters confirmed that the experimental variants with duplicate leaf spraying of *S. kudriavzevii* YSW1 and *S. cerevisiae* YD5 strains are not similar to the leaf-inoculated tobacco plants (Figure 3). Two main groups are formed, the first one combining the three strains applied through foliar spraying and two soil treated *S. kudriavzevii* YSW1 variants, where a higher stimulatory effect was observed. The second group consists of the rest soil drenched yeast strains and twofold leaf-sprayed *S. cerevisiae* YD5 and *S. kudriavzevii* YSW1 strains, where the enhancement was less pronounced (*p* < 0.05).

## 4. Conclusions

It is widely known that leaf chlorophyll content is an important parameter for testing plant status: it can be used as an index of the photosynthetic potential as well as of the plant productivity. The photosynthetic process is one of the first to be disturbed when stress occurs in plants tissues. The data from the periodical observations made on the content of photosynthetic pigments and their ratios, as well as on the intensity of photosynthesis, transpiration, and stomatal conductivity, clearly demonstrated the three studied endophytic yeast strains could be used as growth-promoting bioagents in tobacco plantations. Furthermore, their positive function is not only limited to plant development but they also exert a strong antifungal activity. Plant diseases caused by fungal pathogens are responsible for major crop losses worldwide. Yeast exhibited a high potential to be used as an alternative strategy to manage fungal diseases in tobacco.

## Figures and Tables

**Figure 1 plants-11-00751-f001:**
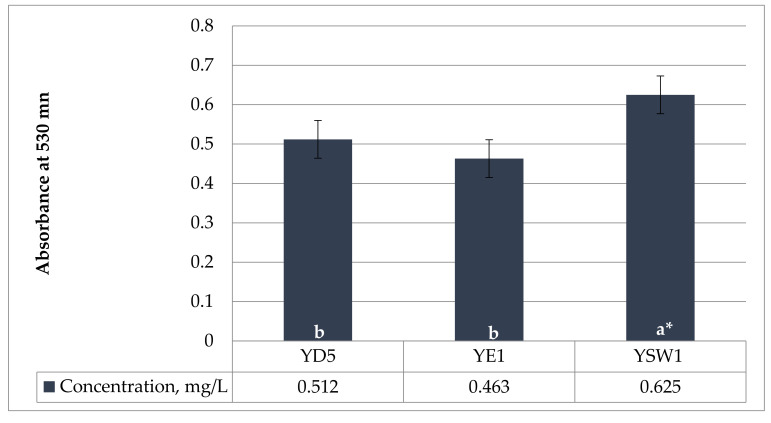
Production of indole-3-acetic acid (IAA) in YPD medium, with 0.1% (*w/v*) L-tryptophan, incubated in a shaker at 30 °C and 150 rpm for 5 days. a and b indicate statistical references, * indicates high significance.

**Figure 2 plants-11-00751-f002:**
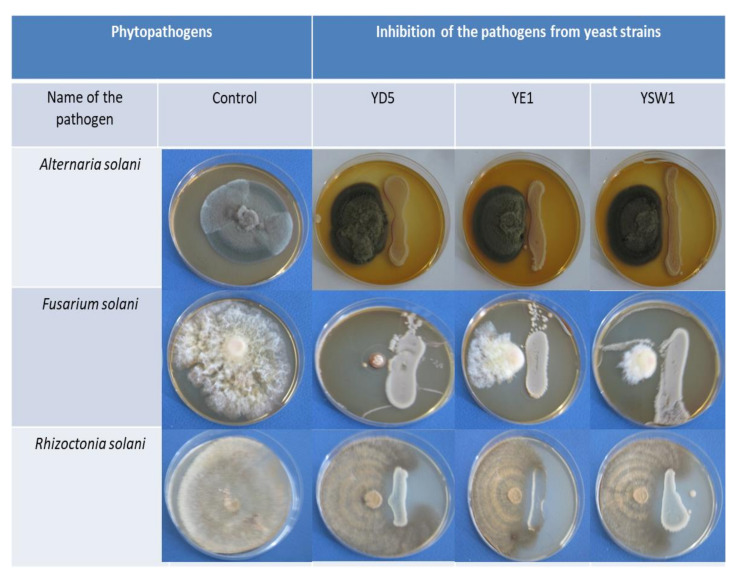
Antimicrobial activities of isolated yeast strains against three phytopathogens: *Alternaria solani pathovar. tobacco, Rhizoctonia solani,* and *Fusarium solani pathovar. phaseoli*.

**Figure 3 plants-11-00751-f003:**
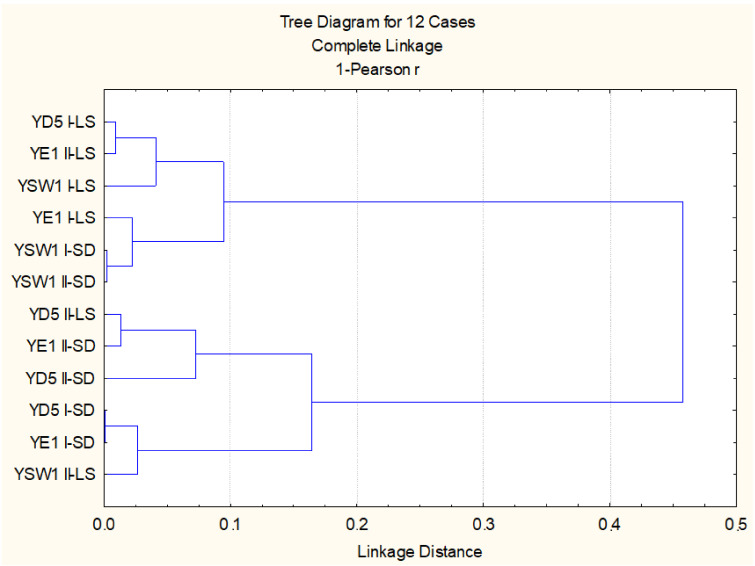
Cluster analysis on the base of all studied biometric and physiological parameters.

**Table 1 plants-11-00751-t001:** Identification and biochemical characterization of isolated strains.

Identification and Biochemical Characterization	Isolated Yeast Strains		
	YD5	YE1	YSW1
**ID**	** *Saccharomyces cerevisiae* **	** *Zygosaccharomyces bailii* **	** *Saccharomyces kudravzevii* **
Level of similarity,%	100	100	100
Accession number in the Gene bank	OL889951	OL904963	OL889954
Phosphate-solubilization index (PSI) as a zone in mm ± SD, *n* = 3	16.60 ± 0.142	27.60 ± 0.19 *	16.00 ± 0.138
Proteolytic activity, measured as zone in mm ± SD, *n* = 3	35.00 ± 0.217 *	16.40 ± 0.152	32.50 ± 0.302
Yellow-orange halo of siderophore producing strains in mm, *n* = 3	+++ *	+++ *	++
NH_3_ production	+++	+++	+++

In case of siderophore activity + shows small halos <10 mm, ++ shows medium diameter of 10–20 mm, whereas +++ shows diameter >20 mm. SD—standard deviation; N—number of replicates. In the case of NH_3_ production, brown-to-orange color development has been considered as a positive result. *—Highly significant.

**Table 2 plants-11-00751-t002:** Percentage of colonization of tobacco plants on day 7 DAT, 14 DAT, and 21 DAT with yeast strains imported by direct inoculation into the soil drench and leaf spraying.

	7th DAT						14th DAT						21st DAT					
	SD			LS			SD			LS			SD			LS		
	Roots	Leaves	Stems	Roots	Leaves	Stems	Roots	Leaves	Stems	Roots	Leaves	Stems	Roots	Leaves	Stems	Roots	Leaves	Stems
Control tobacco plants	0	0	0	0	0	0	0	0	0	0	0	0	0	0	0	0	0	0
Treated with YD5 tobacco plants	66.6 b	33.3 b	16.6 a	33.3 a	83.3 c	75.0 b	85.7 b	100 b *	85.0 b	100.0 c	100.0 b *	100.0 c	100.0 b *	100.0 c	100.0 b *	100.0 b *	100.0 b	100.0 b *
Treated with YE1 tobacco plants	50.0 a	15.0 a	20.0 a	40.0 a	25.0 a	50.0 a *	60.0 a *	75.0 a	66.6 a *	50.0 a	66.0 a	45.0 a	66.6 a	65.0 a	60.0 a	83.3 a	80.0 a *	75.0 a
Treated with YSW1 tobacco plants	75.0 c	45.0 c	50.0 b *	0	66.6 b	75.0 b	75.0 b	70.0 a	85.0 b	75.0 b	75.0 a	75.0 b	66.6 a	85.0 b	60.0 a	100.0 b	100.0 b	86.6 a

Legend: a, b, and c indicate statistical references, * indicates high significance (*p* < 0.05).

**Table 3 plants-11-00751-t003:** Influence of biocontrol yeast on the growth parameters of tobacco plants. Tobacco plants were treated by soil drench (SD) and leaf spraying (LS), and measurements were taken on the 28th DAT (I: first treatment) and the 14th DAT (II: second treatment).

Yeast	Treatment Type	Stem Height ± SD, cm	Root Length ± SD, cm	Leaf Length ± SD, cm	Leaves Biomass ± SD, g
Control	-	21.3 ± 4.21 b	4.80 ± 2.02 a	5.65 ± 3.23 a	3.63 ± 0.86 b
YD5	LS I	23.0 ± 5.48 b	7.25 ± 1.88 b	17.0 ± 3.82 c	7.6 ± 0.55 d
YD5	LS II	28.4 ± 5.32 c	10.5 ± 3.25 c	15.3 ± 3.54 bc	3.7 ± 0.27 b
YD5	SD I	26.5 ± 2.94 c	5.8 ± 2.27 ab	11.2 ± 2.09 b	3.2 ± 0.31 b
YD5	SD II	23.5 ± 3.26 b	8.8 ± 3.33 bc	10.5 ± 2.51 b	4.6 ± 0.88 bc
YE1	LS I	39.5 ± 4.82 e	5.5 ± 2.14 ab	20.0 ± 3.28 d	15.3 ± 2.39 f
YE1	LS II	32.5 ± 3.07 d	7.5 ± 2.63 b	8.6 ± 1.38 ab	10.5 ± 1.81 e
YE1	SD I	38.4 ± 3.85 e	8.0 ± 1.21 bc	13.5 ± 3.99 bc	4.9 ± 0.66 bc
YE1	SD II	26.8 ± 4.66 c	8.5 ± 2.31 bc	17.4 ± 3.87 c	6.0 ± 0.81 c
YSW1	LS I	18.0 ± 4.01 a	8.2 ± 2.19 bc	7.5 ± 2.25 ab	1.5 ± 0.02 a
YSW1	LS II	23.0 ± 4.23 b	7.1 ± 2.70 b	12.5 ± 3.13 b	3.6 ± 0.11 b
YSW1	SD I	37.5 ± 5.64 de	6.2 ± 2.92 ab	14.5 ± 3.40 bc	5.3 ± 0.45 bc
YSW1	SD II	35.0 ± 6.58 d	8.7 ± 2.92 bc	20.7 ± 5.11 d	11.5 ± 1.76 e

Legend: Means with different letters (a, b, c, d, e, and f) in the same column differ at *p* < 0.05 level of probability by an LSD test.

**Table 4 plants-11-00751-t004:** Growth rate of tobacco plants during the experimental period, monitored at 7th, 14th and 28th DAT (single treatment).

Yeast	Treatment Type	Parameter	1–14 DAT Growth Rate, cm	15–28 DAT Growth Rate, cm	Total Growth Rate, cm
Control	No treatment	Mean	7.52 ab	3.26 b	10.78 bc
		SD	5.06	2.48	7.50
		VAR	25.63	615	56.23
		Var%	67.32	76.09	69.56
YD5	LS	Mean	10.86 ab	11.08 a	21.94 ab
		SD	7.89	4.72	12.21
		VAR	62.32	22.29	149.19
		Var%	72.69	42.61	55.67
YD5	SD	Mean	8.5 ab	2.66 b	11.16 bc
		SD	3.95	0.82	4.19
		VAR	15.63	0.67	17.59
		Var%	46.50	30.73	37.58
YE1	LS	Mean	10.42 ab	12.08 a	23.30 a
		SD	5.54	6.36	11.88
		VAR	30.71	40.48	141.08
		Var%	53.18	49.40	50.98
YE1	SD	Mean	10.1 ab	2.80 a	12.90 abc
		SD	5.45	1.37	6.38
		VAR	29.68	1.89	40.66
		Var%	53.94	49.03	49.43
YSW1	LS	Mean	4.90 b	1.48 b	6.38 c
		SD	3.83	0.75	3.95
		VAR	14.68	0.56	15.61
		Var%	78.18	50.65	61.92
YSW1	SD	Mean	14.18 a	4.24 b	18.42 ab
		SD	6.92	1.43	8.06
		VAR	47.91	2.06	64.95
		Var%	48.81	33.83	43.75

a, b, c—Means marked with different letters are significantly different at *p* < 0.05.

**Table 5 plants-11-00751-t005:** Assay for photosynthetic pigments in experimental plants treated by soil drench (SD) and leaf spraying (LS) 14 day after the first treatment.

Yeast Strain	Treatment Type	Concentration ± SD, mg g^−1^ Fresh Weight				Ratio	
		chl a	chl b	chl a + b	car	chl a/b	chl a + b/car
Control	-	4.91 ± 1.3 a	2.86 ± 0.8 a	7.77 ± 2.1 a	1.94 ± 0.4 a	1.72 a	4.02 a
YD5	LS	14.40 ± 2.5 c	8.41 ± 1.7 c	22.81 ± 4.2 d	5.83 ± 1.6 c	1.71 a	3.91 a
YD5	SD	10.95 ± 2.0 b	6.32 ± 1.6 b	17.27 ± 3.6 c	4.12 ± 1.2 b	1.73 a	4.19 a
YE1	LS	11.24 ± 2.1 b	6.31 ± 1.6 b	17.55 ± 3.7 c	4.67 ± 1.2 bc	1.78 a	3.76 a
YE1	SD	8.51 ± 1.8 b	5.13 ± 1.4 ab	13.64 ± 3.2 b	3.46 ± 1.0 b	1.66 a	3.95 a
YSW1	LS	5.69 ± 1.5 a	3.25 ± 1.0 a	8.94 ± 2.5 a	2.14 ± 0.6 a	1.75 a	4.17 a
YSW1	SD	13.40 ± 2.3 c	8.24 ± 1.7 c	21.64 ± 4.0 d	5.13 ± 1.5 c	1.63 a	4.22 a

Legend: Means with different letters (a, b, c, d) in the same column differ at *p* < 0.05 level of probability by LSD test.

**Table 6 plants-11-00751-t006:** An assay for photosynthetic pigments in experimental plants treated by soil drench (SD) and leaf spraying (LS) at 28 days after first treatment and 14 days after the second treatment.

Yeast Strain	Treatment Type	Concentration ± SD, mg g^−1^ Fresh Weight				Ratio	
		chl a	chl b	chl a + b	car	chl a/b	chl a + b/car
Control	-	5.67 ± 1.5 a	3.12 ± 0.9 a	8.79 ± 2.4 a	2.20 ± 0.5 a	1.82 b	3.99 a
YD5	LS I	6.75 ± 1.6 ab	4.15 ± 1.2 ab	10.90 ± 2.8 b	2.48 ± 0.5 ab	1.63 a	4.39 b
YD5	LS II	7.31 ± 1.9 ab	4.32 ± 1.3 ab	11.63 ± 3.2 b	2.97 ± 0.6 ab	1.69 a	3.92 a
YD5	SD I	10.93 ± 2.1 c	6.68 ± 1.7 b	17.62 ± 3.8 d	4.56 ± 1.3 b	1.64 a	3.87 a
YD5	SD II	8.73 ± 1.8 b	5.66 ± 1.5 b	14.39 ± 3.3 c	3.35 ± 1.0 b	1.54 a	4.29 a
YE1	LS I	6.96 ± 1.7 ab	4.40 ± 1.3 ab	11.37 ± 3.0 b	2.56 ± 0.6 ab	1.58 a	4.45 b
YE1	LS II	9.70 ± 1.9 bc	5.91 ± 1.5 b	15.61 ± 3.4 c	3.75 ± 1.0 b	1.64 a	4.17 a
YE1	SD I	8.49 ± 1.7 b	5.14 ± 1.4 ab	13.63 ± 3.1 c	3.04 ± 0.9 ab	1.65 a	4.48 b
YE1	SD II	9.30 ± 1.8 bc	5.52 ± 1.5 b	14.81 ± 3.3 c	3.48 ± 1.0 b	1.69 a	4.26 a
YSW1	LS I	7.01 ± 1.7 ab	4.30 ± 1.4 ab	11.30 ± 3.1 b	2.56 ± 0.6 ab	1.63 a	4.41 b
YSW1	LS II	4.49 ± 1.3 a	2.98 ± 0.8 a	7.47 ± 2.1 a	1.89 ± 0.3 a	1.51 a	3.94 a
YSW1	SD I	9.04 ± 1.8 b	5.95 ± 1.5 b	14.99 ± 3.3 c	3.67 ± 1.1 b	1.52 a	4.08 a
YSW1	SD II	11.10 ± 2.1 c	6.69 ± 1.8 b	17.79 ± 3.9 d	4.17 ± 1.2 b	1.66 a	4.26 a

Legend: Means with different letters (a, b, c, d) in the same column differ at *p* < 0.05 level of probability by an LSD test.

**Table 7 plants-11-00751-t007:** Photosynthesis and transpiration rate in experimental tobacco plants treated by soil drench (SD) and leaf spraying (LS) at 14 days after first treatment.

Yeast	Treatment Type	Intensity of Photosynthesis ± SD, μmol m^−2^ s^−1^	Intensity of Transpiration ± SD, μmol m^−2^ s^−1^	Stomatal Conductance ± SD, μmol m^−2^ s^−1^
Control	-	4.665 ± 2.28 a	0.648 ± 0.04 c	18.98 ± 3.78 c
YD5	LS	9.968 ± 3.31 d	0.648 ± 0.11 c	18.61 ± 3.64 c
YD5	SD	6.983 ± 2.76 b	0.601 ± 0.10 bc	16.83 ± 3.52 bc
YE1	LS	6.850 ± 2.56 b	0.552 ± 0.09 b	16.42 ± 3.49 bc
YE1	SD	6.180 ± 2.39 b	0.644 ± 0.11 c	18.52 ± 3.63 c
YSW1	LS	8.054 ± 2.90 c	0.500 ± 0.08 b	14.31 ± 3.20 b
YSW1	SD	4.050 ± 2.12 a	0.274 ± 0.02 a	7.33 ± 2.81 a

Legend: Means with different letters (a, b, c, d) in the same column differ at *p* < 0.05 level of probability by an LSD test.

**Table 8 plants-11-00751-t008:** Photosynthesis and transpiration rate in experimental tobacco plants: 28th day after first treatment and 14th day after second treatment.

Yeast	Treatment Type	Intensity of Photosynthesis ± SD, μmol m^−2^ s^−1^	Intensity of Transpiration ± SD, μmol m^−2^ s^−1^	Stomatal Conductance ± SD, μmol m^−2^ s^−1^
Control	-	12.086 ± 4.33 b	0.136 ± 0.01 a	4.14 ± 1.69 a
YD5	LS I	9.760 ± 2.30 a	0.377 ± 0.03 bc	11.24 ± 309 d
YD5	LS II	10.154 ± 2.33 ab	0.440 ± 0.03 c	13.84 ± 3.73 e
YD5	SD I	8.568 ± 2.28 a	0.396 ± 0.03 c	12.15 ± 3.33 de
YD5	SD II	10.356 ± 2.38 ab	0.278 ± 0.02 b	8.35 ± 2.91 c
YE1	LS I	11.816 ± 3.14 b	0.418 ± 0.03 c	11.13 ± 3.22 d
YE1	LS II	16.188 ± 4.94 c	0.392 ± 0.03 c	10.43 ± 3.05 d
YE1	SD I	10.575 ± 2.32 ab	0.289 ± 0.02 b	8.54 ± 2.93 c
YE1	SD II	10.388 ± 2.31 ab	0.238 ± 0.02 b	6.96 ± 2.78 b
YSW1	LS I	12.626 ± 2.77 b	0.664 ± 0.04 d	15.65 ± 4.21 f
YSW1	LS II	9.654 ± 2.13 a	0.603 ± 0.04 d	14.31 ± 3.99 f
YSW1	SD I	11.014 ± 2.52 ab	0.226 ± 0.02 ab	5.34 ± 2.65 ab
YSW1	SD II	15.231 ± 4.09 c	0.266 ± 0.02 b	6.34 ± 2.70 b

Legend: Means with different letters (a, b, c, d, e, and f) in the same column differ at *p* < 0.05 level of probability by an LSD test.

## Data Availability

Not applicable.

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
