# Peer review of "Tobacco Plant Growth-Promoting and Antifungal Activities of Three Endophytic Yeast Strains"

_plants, 2022, doi:10.3390/plants11060751_

Round 1

Reviewer 1 Report

The current manuscript "Tobacco Plant Growth-Promoting and Antifungal Activities of Three Endophytic Yeast Strains" The idea seems to be good and the results are promising for publication. However, some comments and modification have to address before publication.

  • The abstract part needs to rewrite in a way to define the exact novelty and originality of your work.
  • All abbreviations used should be mentioned in the place of their first mention followed by an abbreviation and then only the abbreviation is written.
  • Line 21 "YPD"? and line 118 "YMA"
  • Write the source of all mentioned media in international form (Company name, City, Country)
  • During endophytic yeasts isolation, any plants has been used, and mention name of plants with GPS for each plant used.
  • How can you be sure their isolates endophytes and not epiphytes (success of surface sterilization) should be discuss in detail?
  • The authors should replace the symbols of yeast strains with their names.
  • Line 55 "Saccharomyces cerevisiae" replace to italic form.
  • Line 120 replace 25° C by 25 °C, so on.
  • Line 144 " 150 rpm/min" while line 146 " 3000 g for 5 min"?
  • Line 155 NH3 replace by NH3
  • Line 168 replace 2.2.3. by 2.2.4.
  • Line 121 "malt extract medium" while line 169 "Malt Extract (ME)"?
  • Line 103 "Chrome Azurol S (CAS)", line 170 "CAS (Chrome Azurol S)" and line 362 "Chrome Azurol S"?
  • Line 166 add dot after days. Replace Fungi by yeasts.
  • Line 162 "isolated rhizosphere fungi" explain?
  • Line 179 "Fusarium, Rhizoctonia, and Alternaria" replace to italic form without bold.
  • Lines 233-235: please rewrite it.
  • Lines 252, 335, 387, 389, and 421 different fonts' style.
  • Figure or Fig.?
  • Line 357 replace Fe3+ by Fe3+.
  • Line 365 "Supplement figure 1" where?
  • Lines 513-516: please rewrite it.
  • Recent advancements in endophytes for promotion of plant growth are crucial to the agriculture field improvement. Recent papers (for example, https://doi.org/10.3390/plants10050935, https://doi.org/10.1007/978-981-13-8495-0_6). Authors may include this aspect in the discussion section.
  • After identification, each organism used should be named strain not isolate, please check this in manuscript. For example, lines 408, 419 and 427, yeast isolates should be replaced by yeast strains.
  • The authors should clarify the abbreviations in Figures 6 and 7.
  • Reference citing format is not uniform, and the Authors need to follow the desired formats for the Plants.
  • The whole manuscript must be checked to avoid the presentation of same information several times.
  • The text must be checked carefully in order to correct the editing errors.

Author Response

Dear reviewers,

Special thanks for the good comments and suggestions. We appreciate for Editors/Reviewers' warm work earnestly and hope that the correction will meet with approval. 

Point 1. Line 25: please correct to ‘were analyzed’.

Except and corrected: Line 33 were analyzed

Point 2. Lines 27-28: change the tense of this sentence to match with the rest.

Except and corrected Line 34 - The effect of single and double treatment with yeast inoculum on the development and biochemical parameters of tobacco was reported.

Point 3. Line 31-31: In contrast, YSW1 had a better effect when applied as a soil drench.

Except and corrected Line 44-45- had a better effect when applied as a soil 44 drench.

Point 4. Lines 45-52: There are many journal articles of relevance that should be cited here. Otherwise, it seems work with microbial inoculants have been in place only since 2018.

Except and corrected - Line 677, 681, 712, 726, 783

Point 5. Lines 100-105: potential application in agriculture for ? …the initial stages. This sentence is not clear. So are the next few sentences.

Except and corrected Line 108 Saccharomyces cerevisiae was reported to have the potential to inhibit plant pathogens (Ferraz et al. 2019). Among fungal pathogens, chitin-containing pathogens such as Fusarium solani, Rhizoctonia solani, and Alternaria solani induces damage to tobacco such as root and stem rot and leave brown spots (Raza at al., 2013).

Point 6. The abstract part needs to rewrite in a way to define the exact novelty and originality of your work.

Except and of novelty and originality of our work has been underlined with the following sentence:

Line 34-35 - The effect of single and double treatment with yeast inoculum on the 35 development and biochemical parameters of tobacco was reported.

Point 7. All abbreviations used should be mentioned in the place of their first mention followed by an abbreviation and then only the abbreviation is written. Write the source of all mentioned media in international form (Company name, City, Country)

Except and correction were made

Point 8. Line 21 "YPD"? and line 118 "YMA"

Except and correction were made

Line 30-31 yeast malts agar (YMA)

Line 183-184 yeast extract agar 184 (Merck KGaA, Darmstadt, Germany)

  • Point 9. During endophytic yeasts isolation, any plants has been used, and mention name of plants with GPS for each plant used.

Line 147-149. Yeast was isolated from surface-sterilized leaves of strawberry (Fragaria × 147 ananassa) and wheat seeds (Triticum aestivum) grown on the training148 experimental field of the Agricultural University of Plovdiv, spreading on 185 149 ha around the city of Plovdiv, South Bulgaria.

Point 10. How can you be sure their isolates endophytes and not epiphytes (success of surface sterilization) should be discussed in detail? What is superficial disinfection and how was this done?

Corrections Line 149-161 The plant samples were washed 150 with running tap water for one hour to remove dust and debris adhering to 151 them (Hallman et al. 2007). The surface sterilization to remove the adhering microorganisms was done by immersion in a 3% sodium hypochlorite (commercially available) solution for three minutes. Then they were rinsed with 70% ethanol for a minute. The seed and leaves finally were rinsed with deionized sterile distilled water to remove the superficial chemical agents. The sterilized seed and leaf explants were cultured in Petri dishes containing YMA (Himedia Laboratories Pvt., Mumbai, India) supplemented with 100 μg/mL of chloramphenicol. The Petri dishes were sealed with parafilm and incubated at 27°C for 5 days under dark conditions and monitored every day. Yeasts colonies, isolated from each plant explant was subculture on separate YMA plates at room temperature, morphologically analyzed by microscope observation and then identified.

Point 11. What is/are the concentration(s) of Chloramphenicol used in the media? Many bacteria can still grow on chloramphenicol.

Except for the correction

Line 156-157 Petri dishes containing YMA (Himedia Laboratories Pvt., Mumbai, India) supplemented with 100 μg/mL of chloramphenicol.

Point 12. Lines 136-138: The differences in substitution that was obtained in this experimental should be reported as a table.

Except the suggestion and an explanatory figure has been added

Line 903-904 Supplement figure 1. A dendrogram was inferred using the Neighbor-Joining method. This analysis involved 3 nucleotide sequences. There were a total of 1199 positions in the final dataset. Evolutionary analyses were conducted in MEGA X software (Kumar S., Stecher G., Li M., Knyaz C., and Tamura K. 2018. MEGA X: Molecular Evolutionary Genetics Analysis across computing platforms. Molecular Biology and Evolution 35:1547-1549).

Point 13. The authors should replace the symbols of yeast strains with their names.

Except for the correction and the symbols of yeast strains replacement have been done with their names

Lines 237, 269, 409-410, 423-425, 457-458,, 479, 520-522, 541-552, 599,560, 661, 563, 571-573, 594, 631-637, 640-645.

Point 14. Line 55 "Saccharomyces cerevisiae" replace to italic form.

Line 69 - Saccharomyces cerevisiae

Point 15. Line 120 replace 25° C by 25 °C, so on.

Except and correction has been made in the whole text

Line 156-177 27°C for 5 day

Point 16. Line 144 " 150 rpm/min" while line 146 " 3000 g for 5 min"?

Line 187 -  3000 rpm

Point 17. Line 155 NH3 replaced by NH3

Except and corrected Line 197 Assay for NH3 production

Point 18. Line 168 replace 2.2.3. by 2.2.4.

Except and correction has been made

Line 197, 203, 211

Point 19. Line 121 "malt extract medium" while line 169 "Malt Extract (ME)"?

Except and corrected Line 213 yeast malts agar

Point 20. Line 103 "Chrome Azurol S (CAS)", line 170 "CAS (Chrome Azurol S)" and line 362 "Chrome Azurol S"?

Except and corrections has been made

Line 133 Chrome Azurol S (CAS) agar

Line 214 and 455 CAS

Point 21. Line 166 add a dot after days. Replace Fungi with yeasts.

Line 159 Yeasts colonies

Point 23. Line 162 "isolated rhizosphere fungi" explain?

New explanation of isolation and surface sterilization was added

Line 146-161 -Yeast was isolated from surface-sterilized leaves of strawberry (Fragaria × ananassa) and wheat seeds (Triticum aestivum) grown on the training-experimental field of the Agricultural University of Plovdiv, spreading on 185 ha around the city of Plovdiv, South Bulgaria. The plant samples were washed with running tap water for one hour to remove dust and debris adhering to them (Hallman et al. 2007). The surface sterilization to remove the adhering microorganisms was done by immersion in a 3% sodium hypochlorite (commercially available) solution for three minutes. Then they were rinsed with 70% ethanol for a minute. The seed and leaves finally were rinsed with deionized sterile distilled water to remove the superficial chemical agents. The sterilized seed and leaf explants were cultured in Petri dishes containing YMA (Himedia Laboratories Pvt., Mumbai, India) supplemented with 100 μg/mL of chloramphenicol. The Petri dishes were sealed with parafilm and incubated at 27°C for 5 days under dark conditions and monitored every day. Yeasts colonies, isolated from each plant explant was subculture on separate YMA plates at room temperature, morphologically analyzed by microscope observation and then identified.

Point 24. Line 179 "Fusarium, Rhizoctonia, and Alternaria" replace to italic form without bold.

Except and corrections has been made

Line 224- Antimicrobial activity against fungi of the genus Fusarium, Rhizoctonia, and Alternaria

Line 181: Include the pathogen strain (pathovar) tested. Mention the same in Fig.2 as well.

Line 228-Pathogenic fungi were isolated from Solanaceous plants according to Koch

Point 25. What was the concentration of the other antibiotics added? 0.02g ampicillin is mentioned but not for the other two.

Except and corrected Line 264-265 100 μg/ml ampicillin, 100 μg/ml streptomycin and 100 μg/ml tetracycline.

Point 26. Line 218: Where is the formula mentioned in the text?

Except and corrected - the article was incorrectly used.

Line 268-replaced with the correct reference by Petrini and Fisher (1986).

Point 27. Lines 252, 335, 387, 389, and 421 different fonts' style.

Except and corrections have been made

Point 28. Line 118: Monitoring the effect of yeast colonization – was this experiment repeated?

Biometric measurements have been made regularly in 3 repetitions to track the effect of yeast colonization on tobacco plant development.

Point 29. Figure or Fig.?

Figure was chosen and corrected in the whole text Line 380, 503, and 531.

Point 30. Fig 1: How many replicates were tested? N=?. Also the statistical analysis above the bars is missing.

Answer: N=3

Point 30. Line 357 replace Fe3+ by Fe3+.

Except and corrected Line 476 Fe3+.

Point 31. Line 365 "Supplement figure 1" where?

Line 930 Please find the Supplement figure 2. Standard curve of IAA production by yeast strains.

Point 32. Line 375: two-time decrease is not the right way to express. Have a numerical (eg., %) to explain this.

Point 33. Recent advancements in endophytes for promotion of plant growth are crucial to the agriculture field improvement. Recent papers (for example, https://doi.org/10.3390/plants10050935, https://doi.org/10.1007/978-981-13-8495-0_6). Authors may include this aspect in the discussion section.

Except and more recent reference were added Line 684, 691, 722, 736, 793, 818, 822, 858, 861

Point 34. After identification, each organism used should be named strain not isolate, please check this in manuscript. For example, lines 408, 419 and 427, yeast isolates should be replaced by yeast strains.

Except and corrected 2.2.2. Line 198

Lines 26, 254, 269, 457, 546

Point 35. The authors should clarify the abbreviations in Figures 6 and 7. Figure 4: The pictures are very bad and not suitable for publication. The authors should remove this section.

Based on the recommendations of other reviewer Figures 4 was removed and Figure 5-9 were replaced with Tables 2-7 the biometrical and statistical data with calculated SD - standard deviation

Line 556-Table 2

Line 566-Table 3

Line 577 - Table 4

Line 581-Table 5

Line 606-Table 6

Line 609-Table 7

Point 36. Lines 448-455: There is no statistical analysis supporting these findings.

Answer: Suppliment figure 3. Growth rate of tobacco plants during the experimental period, monitored at 7th, 14th Line 925

Line 626-639 When regarding the effect of leaf spraying versus root drench and the effect of single treatment versus duplicate treatment, the statistical evaluation of all studied biometric and physiological parameters highlighted that for Z. bailii YE1 strain the most effective is twofold foliar inoculation, while for S. cerevisiae YD5 is single soil and single leaf treatment, and for S. kudriavzevii YSW1 strain is twofold soil treatment (p<0.05). Cluster analysis based on studied parameters confirmed that the experimental variants with duplicate leaf spraying of S. kudriavzevii YSW1 and S. cerevisiae YD5 strains are not similar to the leaf inoculated tobacco plants (Figure 4). Two main groups are formed, the first one combining the three strains applied through foliar spraying and two soil treated S. kudriavzevii YSW1 variants, where a higher stimulatory effect was observed. The second group consists of the rest soil drenched yeast strains and twofold leaf sprayed S. cerevisiae YD5 and S. kudriavzevii YSW1 strains where the enhancement was less pronounced (p<0.05).

  • Reference citing format is not uniform, and the Authors need to follow the desired formats for the Plants.

Except and corrected

  • The whole manuscript must be checked to avoid the presentation of same information several times.

Except and corrected

  • The text must be checked carefully in order to correct the editing errors.

Except and corrected

Reviewer 2 Report

please see the comments in attach file 

you should update your references it is very old 

Author Response

(The authors gave the same response as above.)

Reviewer 3 Report

This manuscript describes the potential biostimulant and biocontrol of 3 endophytic yeast strains. This manuscript is still in the draft stages, mainly because of the representation of the data. Statistical analysis has not been carried out although the same is mentioned in the text. Needs major revision before it can be accepted for further peer-review.

Below are mentioned the changes required.

Line 14: Typo ‘isolated’

Line 25: please correct to ‘were analyzed’.

Lines 27-28: change the tense of this sentence to match with the rest.

Line 31-31: In contrast, YSW1 had a better effect when applied as a soil drench.

Lines 45-52: There are many journal articles of relevance that should be cited here. Otherwise, it seems work with microbial inoculants have been in place only since 2018.

Line 55: Italicize Saccharomyces cerevisiae. Kindly check the document for such formatting.

Lines 100-105: potential application in agriculture for ? …the initial stages. This sentence is not clear. So are the next few sentences.

Materials and methods:

Line 115: What is superficial disinfection and how was this done?

Line 116: What was the pH of potassium phosphate buffer? And also, how do you macerate tissue samples with continuous shaking?

Lines 118-119: What is/are the concentration(s) of Chloramphenicol used in the media? Many bacteria can still grow on chloramphenicol.

Lines 134-35: Correct the sentence.

Lines 136-138: The differences in substitution that was obtained in this experimental should be reported as a table.

Line 155: NH3? Kindly correct other missed elements too.

Line 166: for 5 days. Fungi…new sentence

Line 173-74: Change the way this sentence is written.

Line 181: Include the pathogen strain (pathovar) tested. Mention the same in Fig.2 as well.

Line 118: Monitoring the effect of yeast colonization – was this experiment repeated?

Line 214: What was the concentration of the other antibiotics added? 0.02g ampicillin is mentioned but not for the other two.

Line 218: Where is the formula mentioned in the text?

Line 245: significant at p<0.05. In some places in the text p-value is mentioned as p>0.05 such as in line 153. Kindly check.

Line 276: Correct the sentence.

Fig 1: How many replicates were tested? N=?. Also the statistical analysis above the bars is missing.

Lines 299-301: Has this been established? If yes, include references. This sentence does not have any weightage otherwise because the next sentence is contradicting.

Line 367: but lack the activity and not luck.

Line 375: two-time decrease is not the right way to express. Have a numerical (eg., %) to explain this.

Lines 408-409. Looks like the last part of the sentence has merged with the next. Correct this. “Tips limited information”?

Lines 211-414: Rewrite these sentences. Nothing is making sense.

Figure 4: The pictures are very bad and not suitable for publication. The authors should remove this section.

Lines 448-455: There is no statistical analysis supporting these findings.

Figure 5: Should be done based on the actual measurements with appropriate statistics. Mentioning percentage values here is not acceptable. Also, the y-axis is not labeled properly. For example, what is the unit of measurement for stem height? Cm, inches? When error bars are in the graph, they should be accompanied by alphabet separation or **.

Although data analysis mentions the use of statistical analysis, there is no mention in the text or the graphs of the results.

The same hold for Figure 6, 7, 8, 9 as well. All the error bars look the same.

Figure 8, 9: What are the units for stomatal conductance, transpiration rate and photosynthesis intensity?

Author Response

(The authors gave the same response as above.)

Round 2

Reviewer 2 Report

The authors have revised their manuscript accordingly.

Therefore I suggest it is now accepted for publication in Plants

Author Response

Dear Reviews,

Thank you for taking the time of giving us a review. We sincerely appreciate all valuable comments and suggestions, which helped us to improve the quality of the article. Our team tried to deliver and match your expectations. The appropriated changes, suggested by the Reviewers, has been introduced to the manuscript. We hope that our manuscript will be acceptable for publication in Plants Journal.

Line 30 and Line 213 Yeast malt agar corrected

Line 90 killer enzymes were changed to mycocins (extracellular proteins) involved in the inhibition of β-glucan synthesis in the cell wall of sensitive strains

Line 134: for C. aloifolium which is a well known - corrected

Line 150: ‘with tap water’ is enough. Remove ‘for one hour’- corrected

Answer: For the control of the sterilization of the explants, dsH2O from the final washing step was antiseptically dripped onto an antibiotic-free YMA. No yeast colonies were detected from the final washing water used for sterilization of the explants.

Line 153 - Removed ‘for one hour.

Line 161: ‘morphologically analyzed using microscopy and then subjected to molecular identification.’

Line 162-168 Added new information- For the control of the sterilization of the explants, dsH2O from the final washing step was antiseptically dripped onto an antibiotic-free YMA. No yeast colonies were detected from the final washing water used for sterilization of the explants. Yeasts colonies, isolated from each plant explant was subculture on separate YMA plates at room temperature, morphologically analyzed using.

Line 167: ‘morphologically analyzed using microscopy and then subjected to  molecular identification is corrected

Line 171 - genomic DNA was..’ corrected

Line 227: Please include the pathovars of Fusarium solaniRhizoctonia solani and Alternaria solani tested in this study. For example., Fusarium solani f. sp. fisi. Because there are many isolates of each of these fungi with different levels of pathogenicity/aggressiveness. Being specific gives more clarification.

Answer- Accepted and corrected on Lines 23-24, 235-236, and 494-495 - Alternaria solani pathovar. tobacco, Rhizoctonia solani, and Fusarium solani pathovar. phaseoli

Line 276: The plants were removed. Corrected is with was

Figure 3: If arithmetic means and SD bars are indicated, they should accompany alphabet separation or ***. Otherwise, the bar graphs are incomplete.

Answer - Line 552-556 -Figure 3 was replaced by Table 2 for a better demonstration of the results from the percentage of colonization of tobacco plants on day 7 DAT, 14 DAT, 21 DAT. Letters a, b, and c showed the differences in the variation of the percentage of colonization, * - means indicates highly significant

Table 2-8: No stat here as well?

Answer:

Line 590-592 Table 4 with statistical reference such as alphabet separation was added

Line 609-612 Table 5 with statistical reference such as alphabet separation was added

Table 4, 5: Data does not look like percentage as mentioned in the legend. They are of concentrations at mg/g fresh weight. This is inconsistent. Again no stat.

Answer – Line 610 (Table 5) and Line 616 (Table 6) percent removed

Table 6, 7: No stat here too.

Answer- Line 641-644 Table 7 and Line 646-649 Table 8 with statistical reference such as alphabet separation was added

Reviewer 3 Report

Most of the corrections mentioned in the previous review has been corrected. But there are still some glaring mistakes especially in the data analysis. Please have these corrected.

Line 30: Should be yeast malt extract. Please have it corrected in the other parts of the text as well.

Line 90: remove ‘killer’. Toxins can be -cidal or -static and try to mention them this way. Killer is not the right word.

Line 134: for C. aloifolium which is a well know…

Line 150: ‘with tap water’ is enough. Remove ‘for one hour’.

Line 161: ‘morphologically analyzed using microscopy and then subjected to molecular identification.’

Line 164: ‘genomic DNA was..’

Line 227: Please include the pathovars of Fusarium solani, Rhizoctonia solani and Alternaria solani tested in this study. For example., Fusarium solani f. sp. fisi. Because there are many isolates of each of these fungi with different levels of pathogenicity/aggressiveness. Being specific gives more clarification.

Line 259: The plants were removed.

Line 267: yeast strains was calculated.

Figure1: The graph has error bars but not statistical reference such as alphabet separation or ***

Figure 3: If arithmetic means and SD bars are indicated, they should accompany alphabet separation or ***. Otherwise, the bar graphs are incomplete.

Table 2: No stat here as well?

Table 4, 5: Data does not look like percentage as mentioned in the legend. They are of concentrations at mg/g fresh weight. This is inconsistent. Again no stat.

Table 6, 7: No stat here too.

Author Response

(The authors gave the same response as above.)
